# Sample complexity and effective dimension for regression on manifolds

**Andrew D. McRae**    **Justin Romberg**    **Mark A. Davenport**
School of Electrical and Computer Engineering
Georgia Institute of Technology
Atlanta, GA 30332
`admcrae@gatech.edu, jrom@ece.gatech.edu, mdav@gatech.edu`

## Abstract

We consider the theory of regression on a manifold using reproducing kernel Hilbert space methods. Manifold models arise in a wide variety of modern machine learning problems, and our goal is to help understand the effectiveness of various implicit and explicit dimensionality-reduction methods that exploit manifold structure. Our first key contribution is to establish a novel nonasymptotic version of the Weyl law from differential geometry. From this we are able to show that certain spaces of smooth functions on a manifold are effectively finite-dimensional, with a complexity that scales according to the manifold dimension rather than any ambient data dimension. Finally, we show that given (potentially noisy) function values taken uniformly at random over a manifold, a kernel regression estimator (derived from the spectral decomposition of the manifold) yields minimax-optimal error bounds that are controlled by the effective dimension.

## 1  Introduction

High-dimensional data is ubiquitous in modern machine learning. Examples include images (2-D and 3-D), document texts, DNA, and neural recordings. In many cases, the number of dimensions in the data is much larger than the number of actual data samples. Traditional statistical methods cannot handle such cases, so researchers have turned to a variety of explicit dimensionality-reduction techniques—which make inference more tractable—and to tools such as neural networks that often implicitly transform the data into a much lower-dimensional feature space. These techniques inherently assume that the data have an *intrinsic* dimension that is much lower than that of the data's original representation. Our goal in this paper is to show that the difficulty of a supervised learning problem depends only on this intrinsic dimension and not on the (potentially much larger) ambient dimension. In particular, we consider the common assumption that the data lie on a low-dimensional *manifold* embedded in Euclidean space (see [1–4] for some of the many example applications).

As an illustration of the kind of results we hope to obtain, we first consider a simple example: a function on the circle $S^1$ (or, equivalently, a periodic function on the real line). Specifically, suppose that we want to estimate a function $f^*$ on the circle from random samples. In general, it is intractable to estimate an arbitrary function from finitely many samples, but it becomes possible if we assume $f^*$ is structured. For example, $f^*$ may exhibit a degree of smoothness, which can be readily characterized via the *Fourier series* for $f^*$. Specifically, recall that we can write $f^*$ as the Fourier series sum $f^*(x) = a_0 + \sum_{\ell \geq 1}(a_\ell \cos(2\pi\ell x) + b_\ell \sin(2\pi\ell x))$. One common notion of smoothness in signal processing is that $f^*$ is *bandlimited*, meaning that this sum can be truncated at some largest frequency $\Omega$. In this case, $f^*$ lies in a subspace of dimension at most $p(\Omega) = 2\lfloor \Omega/2\pi \rfloor + 1$. We know (see, e.g., [5, Chapter 12] or [6]) that we can recover such a function exactly, with high probability, from $n \gtrsim p(\Omega) \log p(\Omega)$ samples placed uniformly at random. If there is measurement noise, the

squared $L_2$ error due to noise scales like $\frac{p(\Omega)}{n}\sigma^2$. In higher dimensions (say, on the torus $T^m$), an $\Omega$-bandlimited function lies in a space of dimension $p(\Omega) = O(\Omega^m)$, and the number of random samples required scales accordingly.

Another model for smoothness is that $f^*$, rather than being bandlimited, has exponentially-decaying frequency components. For example, suppose the Fourier coefficients satisfy $\sum_\ell e^{t\ell^2}(a_\ell^2 + b_\ell^2) < \infty$ for some $t > 0$ (this is roughly equivalent to $f^*$ being the convolution of a Gaussian function with an arbitrary function in $L_2$). The space of such functions is infinite-dimensional, but any function in it can be approximated as $\Omega$-bandlimited to within an error of size $O(e^{-c\Omega^2 t})$, which should enable us to recover a close approximation to $f^*$ from $O(p(\Omega) \log p(\Omega))$ samples.

In this paper, we provide precise analogs of these sample complexity results in the general case of a function on an arbitrary manifold $\mathcal{M}$ with dimension $m$. As on the circle or torus, an $L_2$ function $f(x)$ on a Riemannian manifold has a *spectral decomposition* into modes $u_\ell(x)$ corresponding to vibrational frequencies $\omega_\ell$ for all non-negative integers $\ell$; these modes are the eigenfunctions of the Laplace-Beltrami operator on $\mathcal{M}$. Our first key contribution (described in Theorem 2) is a nonasymptotic version of the *Weyl law* from differential geometry: this states that, for large enough $\Omega$, the set $\mathcal{H}_\Omega^{\mathrm{bl}}$ of $\Omega$-bandlimited functions on $\mathcal{M}$ (functions composed of modes with frequencies below $\Omega$) has dimension $\dim(\mathcal{H}_\Omega^{\mathrm{bl}}) \leq C_m \operatorname{vol}(\mathcal{M}) \Omega^m =: p(\Omega)$. Thus the number of degrees of freedom scales according to the *manifold dimension $m$* rather than a larger ambient dimension.

Our second key contribution is an error bound for recovering functions on $\mathcal{M}$ from randomly-placed samples using kernel regression. We show in Theorem 3 that if we take $n \gtrsim p(\Omega) \log p(\Omega)$ samples of $f^*$, we can recover any $\Omega$-bandlimited function with error

$$\frac{\|\hat{f} - f^*\|_{L_2}^2}{\operatorname{vol}(\mathcal{M})} \lesssim \frac{p(\Omega)}{n}\sigma^2,$$

which is precisely the error rate for parametric regression in a $D(\Omega)$-dimensional space. Our results extend further to approximately-bandlimited functions: for example, if $f^*$ satisfies $\sum_\ell a_\ell^2 e^{t\omega_\ell^2} < \infty$, where $f^* = \sum_\ell a_\ell u_\ell$, then, again with $n \gtrsim p(\Omega) \log p(\Omega)$ samples, we get (Theorem 4)

$$\frac{\|\hat{f} - f^*\|_{L_2}^2}{\operatorname{vol}(\mathcal{M})} \lesssim \frac{p(\Omega)}{n}\sigma^2 + O(e^{-c\Omega^2 t}).$$

Both bounds are minimax optimal in the presence of noise.

These results follow from our Theorem 1, which is a more general result on regression in a reproducing kernel Hilbert space. Theorems 3 and 4 adapt this result to a specific choice of kernel.

The paper is organized as follows. Sections 2 and 3 describe our framework, survey the relevant literature, and compare it to our results. Section 4 contains our main theoretical results. The proofs are in the appendices in the supplementary material. The key technical results are Theorem 1, which is proved via empirical risk minimization and operator concentration inequalities, and Lemma 1 (used to prove Theorem 2), which is proved via heat kernel comparison results on manifolds of bounded curvature.

## 2 Framework and notation

### 2.1 Kernel regression and interpolation

Kernels provide a convenient and popular framework for nonparametric function estimation. They allow us to treat the evaluation of a nonlinear function as a *linear* operator on a Hilbert space, and they give us a computationally feasible way to estimate such a function (which is often in an infinite-dimensional space) from a finite set of samples. Here, we review some of the key ideas that we will need in analyzing kernel methods.

Let $S$ be an arbitrary set, and suppose $k\colon S \times S \to \mathbf{R}$ is a positive definite kernel. Let $\mathcal{H}$ be its associated reproducing kernel Hilbert space (RKHS), characterized by the identity $f(x) = \langle f, k(\cdot, x)\rangle_\mathcal{H}$ for all $f \in \mathcal{H}$ and $x \in S$.

Now, suppose we have $X_1, \ldots, X_n \in S$, $f^* \in \mathcal{H}$ is an unknown function, and we observe $Y_i = f^*(X_i) + \xi_i$ for $i = 1, \ldots, n$, where the $\xi_i$'s represent noise. A common estimator for $f^*$ is the

regularized empirical risk minimizer

$$\hat{f} = \arg\min_{f \in \mathcal{H}} \ \frac{1}{n} \sum_{i=1}^{n} (Y_i - f(X_i))^2 + \alpha \|f\|_{\mathcal{H}}^2, \tag{1}$$

where $\alpha \geq 0$ is a regularization parameter. The solution to the optimization problem (1) is

$$\hat{f}(x) = \sum_{i=1}^{n} a_i k(x, X_i), \tag{2}$$

where $\boldsymbol{a} = (a_1, \ldots, a_n) \in \mathbf{R}^n$ is given by

$$\boldsymbol{a} = (n\alpha \boldsymbol{I}_n + \boldsymbol{K})^{-1} \boldsymbol{Y},$$

where $\boldsymbol{Y} = (Y_1, \ldots, Y_n) \in \mathbf{R}^n$, $\boldsymbol{K}$ is the kernel matrix on $X_1, \ldots, X_n$ defined by $\boldsymbol{K}_{ij} = k(X_i, X_j)$, and $\boldsymbol{I}_n$ is the $n \times n$ identity matrix.

In general, $\hat{f}$ corresponds to a *ridge regression* estimate of $f^*$. The limiting case $\alpha = 0$ can be recast as the problem

$$\hat{f} = \arg\min_{f \in \mathcal{H}} \ \|f\|_{\mathcal{H}} \text{ s.t. } Y_i = f(X_i), \ i = 1, \ldots, n.$$

In this case, if the $X_i$'s are distinct, then $\hat{f}$ interpolates the measured values of $f^*$.

## 2.2   Kernel integral operator and eigenvalue decomposition

A common tool for analyzing kernel interpolation and regression, which will play a central role in our analysis in Section 4, is the eigenvalue decomposition of a kernel's associated integral operator. The integral operator $\mathcal{T}$ is defined for functions $f$ on $S$ by

$$(\mathcal{T}(f))(x) = \int_S k(x, y) f(y) \, d\mu(y),$$

where $\mu$ is a measure on $S$. Under certain assumptions[1] on $S$, $\mu$, and $k$, $\mathcal{T}$ is a well-defined operator on $L_2(S)$, is compact and positive definite with respect to the $L_2$ inner product, and has eigenvalue decomposition

$$\mathcal{T}(f) = \sum_{\ell=1}^{\infty} t_\ell \langle f, v_\ell \rangle_{L_2} v_\ell, \ f \in L_2(S),$$

where the eigenvalues $\{t_\ell\}$ are arranged in decreasing order and converge to 0, and the eigenfunctions $\{v_\ell\}$ are an orthonormal basis for $L_2(S)$. We also have $k(x, y) = \sum_{\ell=1}^{\infty} t_\ell v_\ell(x) v_\ell(y)$, where the convergence is uniform and in $L_2$.

This eigendecomposition plays an important role in characterizing the RKHS $\mathcal{H}$ associated with the kernel $k$. Combining this expression for $k$ with the identity $\langle f, k(\cdot, x) \rangle_{\mathcal{H}} = f(x)$, we can derive the fact that, for all $f, g \in \mathcal{H}$,

$$\langle f, g \rangle_{\mathcal{H}} = \sum_{\ell=1}^{\infty} \frac{\langle f, v_\ell \rangle_{L_2} \langle g, v_\ell \rangle_{L_2}}{t_\ell}.$$

This implies that $\langle f, g \rangle_{L_2} = \langle \mathcal{T}^{1/2}(f), \mathcal{T}^{1/2}(g) \rangle_{\mathcal{H}}$ for all $f, g \in L_2(S)$. Thus $\mathcal{T}^{1/2}$ is an isometry from $L_2(S)$ to $\mathcal{H}$, and so for any $f \in \mathcal{H}$, we can write $f = \mathcal{T}^{1/2}(f_0)$, where $\|f_0\|_{L_2} = \|f\|_{\mathcal{H}}$. This implies that, for any $p \geq 1$, the projection of $f$ onto $(\text{span}\{v_1, \ldots, v_p\})^{\perp}$ has $L_2$ norm at most $\sqrt{t_{p+1}} \|f\|_{\mathcal{H}}$. Hence the decay of the eigenvalues $\{t_\ell\}$ of $\mathcal{T}$ characterizes the "effective dimension" of $\mathcal{H}$ in $L_2$, which will be a fundamental building block for our analysis.

## 2.3 Spectral decomposition of a manifold and related kernels

We now turn to our specific problem of regression on a manifold, considering how an RKHS framework can help us. The book [8] is an excellent reference for the material in this section.

A smooth, compact Riemannian manifold $\mathcal{M}$ (without boundary) can be analyzed via the *spectral decomposition* of its Laplace-Beltrami operator $\Delta_{\mathcal{M}}$ (we will often call it the Laplacian for short). This operator is defined as $\Delta_{\mathcal{M}} f \coloneqq -\operatorname{div}(\nabla f)$. In $\mathbf{R}^m$, it is simply the operator $-\sum_{i=1}^{m} \frac{\partial^2}{\partial x_i^2}$. The Laplacian can be diagonalized as

$$\Delta_{\mathcal{M}} f = \sum_{\ell=0}^{\infty} \lambda_\ell \langle f, u_\ell \rangle_{L_2} u_\ell,$$

where $0 = \lambda_0 < \lambda_1 \leq \lambda_2 \leq \cdots$, the sequence $\lambda_\ell \to \infty$ as $\ell \to \infty$, and $\{u_\ell\}$ is an orthonormal basis for $L_2(\mathcal{M})$ (all integrals are with respect to the standard volume measure on $\mathcal{M}$).

The eigenvalues $\{\lambda_\ell\}$ are the squared resonant frequencies of $\mathcal{M}$, and the eigenfunctions $\{u_\ell\}$ are the vibrating modes, since solutions to the wave equation $f_{tt} + \Delta_{\mathcal{M}} f = 0$ on $\mathcal{M}$ have the form

$$f(t,x) = \sum_{\ell=0}^{\infty} (a_\ell \sin \sqrt{\lambda_\ell} t + b_\ell \cos \sqrt{\lambda_\ell} t) u_\ell(x).$$

The classical Weyl law (e.g., [8, p. 9]) says that, if $\mathcal{M}$ has dimension $m$, then, asymptotically,

$$|\{\ell : \lambda_\ell \leq \lambda\}| \sim c_m \operatorname{vol}(\mathcal{M}) \lambda^{m/2}$$

as $\lambda \to \infty$, where $c_m = (2\pi)^{-m} V_m$, with $V_m$ denoting the volume of the unit ball in $\mathbf{R}^m$.

Using the spectral decomposition of the Laplacian, any number of kernels can be defined by

$$k(x,y) = \sum_{\ell=0}^{\infty} g(\lambda_\ell) u_\ell(x) u_\ell(y)$$

for some function $g$. With this construction, the integral operator of $k$ has eigenvalue decomposition $\mathcal{T}(f) = \sum_{\ell \geq 0} g(\lambda_\ell) \langle f, u_\ell \rangle_{L_2} u_\ell$, hence, per Section 2.2, $\|f\|_{\mathcal{H}}^2 = \sum_{\ell \geq 0} \langle f, u_\ell \rangle^2 / g(\lambda_\ell)$.

Our results could, in principle, apply to many kernels with the above form, but we will primarily consider *bandlimited kernels* and the *heat kernel*. The bandlimited kernel with bandlimit $\Omega > 0$ is

$$k_\Omega^{\mathrm{bl}}(x,y) = \sum_{\lambda_\ell \leq \Omega^2} u_\ell(x) u_\ell(y),$$

which is the reproducing kernel of the space of bandlimited functions on $\mathcal{M}$:

$$\mathcal{H}_\Omega^{\mathrm{bl}} = \left\{ f \in L_2(\mathcal{M}) : f \in \operatorname{span}\{u_\ell : \lambda_\ell \leq \Omega^2\} \right\}$$

with $\|f\|_{\mathcal{H}_\Omega^{\mathrm{bl}}} = \|f\|_{L_2}$ for $f \in \mathcal{H}_\Omega^{\mathrm{bl}}$. The heat kernel is a natural counterpart to the common Gaussian radial basis function on $\mathbf{R}^m$. Detailed treatments can be found in [8, 9]. We will define it for $t > 0$ as

$$k_t^{\mathrm{h}}(x,y) = \sum_{\ell=0}^{\infty} e^{-\lambda_\ell t/2} u_\ell(x) u_\ell(y).$$

Its corresponding RKHS is

$$\mathcal{H}_t^{\mathrm{h}} = \left\{ f \in L_2(\mathcal{M}) : \|f\|_{\mathcal{H}_t^{\mathrm{h}}}^2 = \sum_{\ell=0}^{\infty} e^{\lambda_\ell t/2} \langle f, u_\ell \rangle_{L_2}^2 < \infty \right\}.$$

The heat kernel $k_t^{\mathrm{h}}$ gets its name from the fact that it is the fundamental solution to the heat equation $f_t + \frac{1}{2}\Delta_{\mathcal{M}} f = 0$ on $\mathcal{M}$. The heat kernel on $\mathbf{R}^m$ is $k_t^{\mathrm{h}}(x,y) = \frac{1}{(2\pi t)^{m/2}} e^{-\|x-y\|^2/2t}$.

# 3 Related work

## 3.1 Dimensionality reduction and low-dimensional structure

There is an extensive literature on the use of low-dimensional manifold structure in machine learning. Perhaps most prominently, nonlinear dimensionality-reduction techniques that exploit manifold structure have been developed, such as [10–14]. More recently, there has been explicit inclusion of manifold models into neural network architectures [15–18]. However, none of this research provides nonasymptotic performance guarantees.

On the other hand, the field of high-dimensional statistics provides many theoretical guarantees for low-dimensional data models. For example, there are extensive bodies of theory for models such as sparsity [19, 20] and low-rank structure [21]. One can view low-dimensional manifold models as a more powerful generalization of such structures. One interesting work that bridges the gap between manifold models and high-dimensional statistics is [22], which is another explicit dimensionality-reduction technique. Another similar line of work is the study of algebraic variety models (e.g., [23]), which are also nonlinear and low-dimensional.

While the great success of the many implicit and explicit dimensionality-reducing methods provides empirical evidence for the possibility of exploiting manifold structure, there are still very large gaps in our theoretical understanding of when and why these methods can be effective.

## 3.2 Manifold regression and kernels

Regression on manifold domains has been explored in a number of previous works. The closely-related problem of density estimation is considered in [24, 25]. Particularly relevant to our paper, [24] uses the same bandlimited kernel and heat kernel that we highlight (and it analyzes the spectral decomposition of these kernels via the asymptotic Weyl law). It is primarily interested in the power of the error rate that can be obtained by assuming the function (density) of interest has a certain number of derivatives; in particular, it shows that $\|\hat{f} - f^*\|_{L_2}^2 \lesssim n^{-2s/(m+2s)}$ if $f$ has $s$ bounded derivatives. Both works, like ours, assume explicit knowledge of the manifold.

Perhaps more relevant to practical applications, [26] seeks to provide a manifold-agnostic algorithm via local linear approximations to the data manifold; however, it is also primarily interested in asymptotic error rates. The paper [27] examines related methods asymptotically in more detail. Another manifold-agnostic method similar in spirit to ours is that of [28], who consider kernel estimation with (Euclidean) Gaussian radial basis functions. They obtain the optimal $n^{-2s/(m+2s)}$ rate for $s$-smooth regression functions; however, their assumptions are quite different from ours in that their regression functions must have *smooth extensions* to (a neighborhood in) the embedding space. Similarly, [29] obtain the optimal rate for functions that are $s$-smooth (in the manifold calculus, similarly to our assumptions) using a neural-network–type architecture. However, they implicitly assume that the manifold is $C^\infty$-embedded in Euclidean space.

In [30, 31], the authors explore Gaussian process models (which are closely related to kernel methods) on a manifold.

The error rate $\|\hat{f} - f^*\|_{L_2}^2 \lesssim n^{-2s/(m+2s)}$ is standard (and minimax optimal) in nonparametric statistics. However, our function model and results are quite different in nature. The regression functions we consider are *infinitely* smooth, and we show that the estimation of these functions is much like a *finite-dimensional* regression problem; not only do we get an $n^{-1}$ error rate (as we do when we take $s \to \infty$ above), but the constant in front of this rate and the minimum number of samples needed are proportional to the finite effective dimension.

Finally, we also note that the idea of using a kernel that can be expressed in terms of the spectral decomposition of a manifold's Laplacian also has precedent. In addition to [24], the paper [32] suggests using such kernels for interpolation in Sobolev spaces on a manifold.

## 3.3 General kernel interpolation and regression

Regression is a strict superset of interpolation; interpolation typically assumes that we sample function values exactly (i.e., there is no noise), while regression allows for (and often assumes) noise.

There is a substantial literature on the use of a kernel for interpolation of functions in an RKHS (often, in this literature, referred to as the "native space" of the kernel). A fairly comprehensive survey can be found in [33]. Distinct from our work, most of this literature considers *deterministic* samples of the function of interest. Given (deterministic) sample locations $\{X_1, \ldots, X_n\} \subset S$, results in this literature tend to have the form $\|\hat{f} - f^*\|_\infty \leq g(h_X)\|f^*\|_{\mathcal{H}}$, where $h_X = \max_{x \in S} \min_{i \in \{1,\ldots,n\}} d(x, X_i)$, and $g(h)$ is a function that decreases to 0 as $h \to 0$ at a rate that depends on the properties of the kernel $k$ (typically as a power or exponentially). Some recent work applying kernel interpolation theory to manifolds is [34–36].

Much of the literature on (noisy) RKHS regression primarily considers the case when the eigenvalues of the integral operator (described in Section 2.1) decay as $t_\ell \lesssim \ell^{-b}$. In [37–39], it is shown that the minimax optimal error rate is $\|f^* - \hat{f}\|_{L_2}^2 \lesssim n^{-b/(b+1)}$. Many other recent papers have explored this rate of convergence in a variety of settings [40–43]. Several of these include more general spectral regularization algorithms, suggested by [44]. Some interesting recent extensions consider a variety of algorithms that may be more practical for large-data situations. These include iterative methods [45–47] and distributed algorithms [48–50].

Another set of results (which are the most similar to ours) uses a regularized effective dimension $p_\alpha = \sum_\ell \frac{t_\ell}{\alpha + t_\ell}$, where $\alpha$ is the regularization parameter. This is considered in [51] and greatly refined in [52]. Variations on these results can be found in [53]. See Section 4.1 for further discussion and comparison to our results. The earlier report [54] resembles our work in its analysis of truncated operators. We note that in the case of power-law eigenvalue decay, these results (and ours) recover the $n^{-b/(b+1)}$ error rate.

It is interesting to note that the squared error rate $n^{-b/(b+1)}$ can recover the standard rate for regression of $s$-smooth functions on manifolds. The Sobolev space of order $s$ is the RKHS of the kernel $\sum_\ell (1 + \lambda_\ell)^{-s} u_\ell(x) u_\ell(y)$. By the Weyl law, its eigenvalues decay according to $t_\ell \approx \ell^{-2s/m}$; plugging $2s/m$ in for $b$ recovers the standard rate $n^{-2s/(m+2s)}$.

# 4 Main theoretical results

## 4.1 Dimensionality in RKHS regression

Here we present our main results for general regression and interpolation in an RKHS. Our results also apply to the slightly more general setting of learning in an arbitrary Hilbert space (see, e.g., [52]), but we do not explore this here. We continue to use the notation established in Sections 2.1 and 2.2, and we further assume that $\mu(S) = 1$ (since $\mu$ is finite, we can always obtain this by a rescaling). We assume that the function samples we take are uniformly distributed on $S$:

**Assumption 1.** The sample locations $X_1, \ldots, X_n$ are i.i.d. according to $\mu$.

Since $\mathcal{H}$ is, in general, infinite-dimensional, there is typically no hope of recovering an arbitrary $f^* \in \mathcal{H}$ to within a small error in $\mathcal{H}$-norm from a finite number of measurements. However, the discussion in Section 2.2 suggests a more feasible goal. Since any set of functions bounded in $\mathcal{H}$-norm can be approximated within an arbitrarily small $L_2$ error in a finite-dimensional subspace of $L_2$, as long as the number of measurements is proportional to this loosely-defined "effective dimension" of $\mathcal{H}$, we have hope of recovering $f^*$ accurately in an $L_2$ sense.

Let $p > 0$ be a fixed integer dimension. Let $G = \text{span}\{v_1, \ldots, v_p\} \subset \mathcal{H} \cap L_2(S)$, and let $G^\perp$ be its orthogonal complement in $L_2(S)$ and $\mathcal{H}$. We denote by $\mathcal{T}_G$ and $\mathcal{T}_{G^\perp}$ the restrictions of $\mathcal{T}$ onto $G$ and $G^\perp$, respectively. We make the following assumptions on the eigenvalues and eigenfunctions of $\mathcal{T}$:

**Assumption 2.** For some constants $K_p$ and $R_p$, we have $\sum_{\ell=1}^p v_\ell^2(x) \leq K_p$ and $\sum_{\ell=p+1}^\infty t_\ell v_\ell^2(x) \leq R_p$ for almost every $x \in S$.

This says that the energy of the eigenfunctions of $\mathcal{T}$ is reasonably spread out over the domain $S$—for the basis $\{v_1, \ldots, v_p\}$, this is a type of incoherence assumption. If the eigenfunctions are well-behaved, we can expect $K_p \approx p$ and $R_p \approx \text{tr}\,\mathcal{T}_{G^\perp}$. This holds in our original example of the Fourier series on the circle, since the sinusoid basis functions are bounded by an absolute constant. Our "pointwise" Weyl law in Theorem 2 shows that we have similar behavior for the

spectral decomposition of a manifold. Note that $K_p$ in Assumption 2 is identical to the quantity $K(p)$ in [6], which uses similar methods to handle a much simpler problem.

**Assumption 3.** For some $\gamma, \gamma' \geq 0$, we have $\frac{\operatorname{tr} \mathcal{T}_{G^\perp}}{t_{p+1}} \leq \gamma p$ and $\frac{R_p}{t_{p+1}} \leq \gamma' K_p$.

This assumption greatly simplifies the notation of our results and is always true with an appropriate choice of $\gamma$ and $\gamma'$. $\gamma$ is often small when $t_{p+1}$ is in the decaying "tail" of eigenvalues. If the eigenvalues decay like $t_\ell \approx \ell^{-b}$, we can take $\gamma \approx (b-1)^{-1}$. Note that a similar assumption appears in [55]. If $K_p \approx p$ and $R_p \approx \operatorname{tr} \mathcal{T}_{G^\perp}$, then $\gamma \approx \gamma'$.

With these assumptions in place, we can state our main theorem for RKHS regression:

**Theorem 1.** *Suppose Assumptions 1 to 3 hold. Let $\delta \in (0,1)$. If*

$$n \geq (7 \vee 3\gamma') K_p \log \frac{(2 \vee 4\gamma) p}{\delta},$$

*then the following hold for the kernel estimate $\hat{f}$ with regularization parameter $\alpha \geq 0$:*

1. *If there is no noise, that is, $Y_i = f^*(X_i)$ for each $i$, then, with probability at least $1 - \delta$, uniformly in $f^*$,*

$$\|\hat{f} - f^*\|_{L_2} \leq (\sqrt{2\alpha} + 6\sqrt{t_{p+1}}) \|f^*\|_{\mathcal{H}}.$$

2. *Now suppose that $Y_i = f(X_i) + \xi_i$, where the $\xi_i$'s are i.i.d., zero-mean, sub-exponential random variables with variance $\sigma^2$ and are independent of the $X_i$'s. If we additionally have*

$$\frac{n}{\log^2 n} \geq C(1 \vee \gamma') \frac{K_p}{p} \frac{\|\xi\|_{\psi_1}^2}{\sigma^2},$$

*where $C$ is a universal constant, and $\alpha \geq 54 t_{p+1}$, then, with probability at least $1 - 2\delta$, uniformly in $f^*$,*

$$\|\hat{f} - f^*\|_{L_2} \leq (\sqrt{2\alpha} + 6\sqrt{t_{p+1}}) \|f^*\|_{\mathcal{H}} + 4\left(1 + \frac{\sqrt{\gamma}}{8}\right) \frac{\sqrt{p} + 2\sqrt{\log 4/\delta}}{\sqrt{n}} \sigma.$$

Our results guarantee an $L_2$ recovery error bounded by two terms: (1) a "bias" depending on the next tail eigenvalue $t_{p+1}$ and the regularization coefficient $\alpha$, and (2) a "variance" term that behaves similarly to the error found in $p$-dimensional regression. When $K_p \approx p$, this result yields the $n \gtrsim p \log p$ sample complexity that we expect. If $\mathcal{H}$ is, in fact, $p$-dimensional (which our framework can handle with $t_\ell = 0$ for $\ell > p$), this result recovers standard $p$-dimensional regression bounds such as those in [6].

We assume i.i.d. noise for simplicity, but our result could easily be extended beyond this case. Note that if the noise is Gaussian, the ratio $\|\xi\|_{\psi_1}^2 / \sigma^2$ is an absolute constant.

For interpolation ($\alpha = 0$) in the noiseless case, this theorem yields $\|\hat{f} - f^*\|_{L_2} \leq 6\sqrt{t_{p+1}} \|f^*\|_{\mathcal{H}}$. In the noisy case, the lower bound on $\alpha$ can be relaxed to get a result with worse constants. We obtain qualitatively similar results whenever $\alpha \gtrsim t_{p+1}$. The assumptions and results of [52] (specialized to our setting) are comparable to Theorem 1 when we set $\alpha \approx t_{p+1}$. However, our results have the advantage of applying even in infinite-dimensional settings with no regularization: the regularized effective dimension $p_\alpha = \sum_\ell \frac{t_\ell}{\alpha + t_\ell}$ from their work would be infinite if $\alpha = 0$.

Although we do not explore it here, we note that one could generalize our approach to the case where the sampling measure differs from that under which the $L_2$ norm is calculated. We could simply bound the ratio (Radon-Nikodym derivative) between the two measures, or we could perform leverage-score sampling to mitigate the need for bounding the eigenfunctions (see, e.g., [55] for similar ideas).

In the presence of noise, Theorem 1 is minimax optimal over the set $\{f \in \mathcal{H} : \|f\|_{\mathcal{H}} \leq r\}$ for any $r > 0$ if $p$ is chosen so that $\frac{p}{n}\sigma^2 \approx t_{p+1}r^2$. In this case,

$$\left\{f \in \operatorname{span}\{v_1, \ldots, v_p\} : \|f\|_{L_2} \lesssim \sqrt{\frac{p}{n}}\sigma\right\} \subset \{f \in \mathcal{H} : \|f\|_{\mathcal{H}} \leq r\},$$

and the minimax rate (with, say, Gaussian noise) over the left-hand set is well-known to be $\sqrt{\frac{p}{n}}\sigma$.

## 4.2 Manifold function estimation

We now describe how we can leverage Theorem 1 to establish sample complexity bounds for regression on a manifold. Suppose, again, that $\mathcal{M}$ is an $m$-dimensional smooth, compact Riemannian manifold. To study the eigenvalues and eigenfunctions of the Laplacian $\Delta_{\mathcal{M}}$, we consider the heat kernel $k_t^{\mathrm{h}}$. Our key tool is the following fact:

**Lemma 1.** *Let $\epsilon \in (0, 2/3)$. Suppose the sectional curvature of $\mathcal{M}$ is bounded above by $\kappa$. For $t \leq \frac{6\epsilon}{(m-1)^2 \kappa}$ and all $x \in \mathcal{M}$,*

$$k_t^{\mathrm{h}}(x, x) \leq \frac{1 + \epsilon}{(2\pi t)^{m/2}}.$$

This is a precise quantification of the well-known asymptotic behavior of the heat kernel as $t \to 0$ (see, e.g., [8, Section VI]). It is derived in Appendix B from a novel set of more general upper and lower bounds for the heat kernel on a manifold of bounded curvature; we note that these may be of independent interest.

Our nonasymptotic Weyl law is a simple consequence of Lemma 1:

**Theorem 2.** *If $\mathcal{M}$ has sectional curvature bounded above by $\kappa$, and $\epsilon \in (0, 2/3)$, then, for all $x \in \mathcal{M}$ and $\lambda \geq \frac{m(m-1)^2 \kappa}{6\epsilon}$,*

$$N_x(\lambda) := \sum_{\lambda_\ell \leq \lambda} u_\ell^2(x) \leq \frac{2(1 + \epsilon)\sqrt{m}}{(2\pi)^m} V_m \lambda^{m/2}.$$

With appropriate rescaling by $\mathrm{vol}(\mathcal{M})$, this gives us a bound on the constant $K_p$ from Section 4.1. Since this result bounds the eigenfunctions, it is a type of "local Weyl law" (see, e.g., [56]). Integrating this result over $\mathcal{M}$ gives a nonasymptotic version of the traditional Weyl law. Our bound is within the modest factor $2(1 + \epsilon)\sqrt{m}$ of the optimal asymptotic law. For simplicity, we will take $\epsilon = 1/2$ in what follows, but slightly better constants could be obtained with smaller $\epsilon$.

The following result for the finite-dimensional bandlimited kernel is a straightforward consequence of Theorems 1 and 2:

**Theorem 3.** *Suppose the sectional curvature of $\mathcal{M}$ is bounded above by $\kappa$. Let $\Omega^2 \geq \frac{m(m-1)^2 \kappa}{3}$, and suppose $f^* \in \mathcal{H}_\Omega^{\mathrm{bl}}$. Let $\hat{f}$ be the kernel regression estimate with kernel $k_\Omega^{\mathrm{bl}}$.[2]*

*Let $\delta \in (0, 1)$, and suppose $n \geq 7p \log \frac{2p}{\delta}$, where*

$$p = p(\Omega) := \frac{3\sqrt{m} V_m}{(2\pi)^m} \mathrm{vol}(\mathcal{M}) \Omega^m. \tag{3}$$

*Under the same noise assumptions as in Theorem 1, if $\frac{n}{\log^2 n} \geq C \|\xi\|_{\psi_1}^2 / \sigma^2$, then, with probability at least $1 - 2\delta$, uniformly in $f^*$,*

$$\frac{\|\hat{f} - f^*\|_{L_2}}{\sqrt{\mathrm{vol}(\mathcal{M})}} \leq 4 \frac{\sqrt{p} + 2\sqrt{\log 4/\delta}}{\sqrt{n}} \sigma.$$

To analyze the heat kernel, which has an infinite number of nonzero eigenvalues, we need the following additional corollary of Lemma 1, which will let us bound the constant $R_p$ from Section 4.1:

**Lemma 2.** *For $\epsilon \in (0, 2/3), t \leq \frac{6\epsilon}{(m-1)^2 \kappa}, \lambda \geq m/t$, and all $x \in \mathcal{M}$,*

$$\sum_{\lambda_\ell \geq \lambda} e^{-\lambda_\ell t/2} u_\ell^2(x) \leq e^{-\lambda t/2} \frac{2(1 + \epsilon)\sqrt{m}}{(2\pi)^m} V_m \lambda^{m/2}.$$

From this, we obtain the following result:

**Theorem 4.** *Suppose the sectional curvature of $\mathcal{M}$ is bounded above by $\kappa$. Let $t \leq \frac{3}{(m-1)^2\kappa}$, and suppose $f^* \in \mathcal{H}_t^{\mathrm{h}}$. Fix $\Omega^2 \geq m/t$, and let $\hat{f}$ be the kernel regression estimate of $f^*$ with kernel $k_t^{\mathrm{h}}$ and regularization parameter $\alpha \geq 54 \frac{e^{-\Omega^2 t/2}}{\mathrm{vol}(\mathcal{M})}$.*

*Let $\delta \in (0,1)$, and suppose $n \geq 7p \log \frac{4p}{\delta}$, with $p$ defined as in (3).*

*Under the same noise assumptions as in Theorem 1, if $\frac{n}{\log^2 n} \geq C \|\xi\|_{\psi_1}^2 / \sigma^2$, then, with probability at least $1 - 2\delta$, uniformly in $f^*$,*

$$\frac{\|\hat{f} - f^*\|_{L_2}}{\sqrt{\mathrm{vol}(\mathcal{M})}} \leq \left( \sqrt{2\alpha} + 6\sqrt{\frac{e^{-\Omega^2 t/2}}{\mathrm{vol}(\mathcal{M})}} \right) \|f^*\|_{\mathcal{H}_t^{\mathrm{h}}} + \frac{9}{2} \frac{\sqrt{p} + 2\sqrt{\log 4/\delta}}{\sqrt{n}} \sigma.$$

These results illustrate how we can exploit the effective finite dimension of spaces of smooth functions on manifolds in regression. This function space dimension (and hence the sample complexity of regression) grows exponentially in the *manifold dimension*, rather than in the larger ambient data dimension, if $\mathcal{M}$ is embedded in a higher-dimensional space. In practice, the true bandlimited or heat kernels may be difficult to compute. It is an interesting open question whether we can obtain similar results for manifold-agnostic algorithms (the work of [26], although it does not apply to our function classes, is an interesting potential starting point).

As discussed in Section 4.1, our general regression result Theorem 1 is similar to prior results [51, 52], but it has the advantage of applying even without regularization in the noiseless case. However, we note that one could obtain results in many ways comparable (minus this advantage) to Theorems 3 and 4 by plugging Theorem 2 and Lemma 2 into those previous regression results. We could not do this with classical power-law results such as [37–39], since our eigenvalue decay is *exponential* rather than power-law.

Since the (classical) Weyl law also *lower* bounds the complexity of spaces of bandlimited functions, then, as discussed in Section 4.1, Theorems 3 and 4 (for the optimally chosen value of $\Omega$) are minimax optimal when there is noise. Furthermore, the requirement $n \gtrsim p \log p$ is necessary in general: if we consider the torus $T^m$, recovering arbitrary $\Omega$-bandlimited functions requires every point on $T^m$ to be within distance $O(1/\Omega)$ of a sample point; considering a uniform grid on $T^m$ and a coupon collector argument makes it clear that $n \gtrsim O(\Omega^m \log \Omega^m)$ randomly sampled points are required.

As mentioned in Section 4.1 for general kernel learning, these results could be extended to consider nonuniform sampling over the manifold.

There are also some very interesting connections between kernel methods and neural networks. The recent works [57, 58] show that trained multi-layer neural networks approach, in the infinite-width limit, a kernel regression function with a "neural tangent kernel" that depends on the initialization distribution of the weights and the network architecture. This follows literature on the connections between Gaussian processes (closely related to kernel methods) and wide neural networks (see, e.g., [59, 60]). It would be very interesting to explore any potential connections between these and the kernels considered in this paper, which are derived from a manifold's spectral decomposition.

## Broader Impact

The results in this paper further illuminate the role of low-dimensional structure in machine learning algorithms. An improved theoretical understanding of the performance of these algorithms is increasingly important as tools from machine learning become ever-more-widely adopted in a range of applications with significant societal implications. Although, in general, there are well-known ethical issues that can arise from inherent biases in the way data are sampled and presented to regression and classification algorithms, we do not have reason to believe that the methods presented in this paper would either enhance or diminish these issues. Our analysis is abstract and, for better or for worse, assumes a completely neutral sampling model (uniform over a manifold).

## Acknowledgments and Disclosure of Funding

This work was supported, in part, by National Science Foundation grant CCF-1350616, a gift from the Alfred P. Sloan Foundation, and the Georgia Tech ARC-TRIAD student fellowship.

## Footnotes

[1] E.g., $S$ is a compact metric space; $\mu$ is strictly positive, finite, and Borel; and $k$ is continuous [7].

[2]The calculation of this estimate is somewhat different than usual, since the rank of the kernel matrix $\boldsymbol{K}$ is at most the dimension of $\mathcal{H}_\Omega^{\mathrm{bl}}$. We do not use regularization, but we use the Moore-Penrose pseudoinverse of $\boldsymbol{K}$ instead of $\boldsymbol{K}^{-1}$.

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
