[Supplementary Material]

# A  Proof of general RKHS results (Theorem 1)

We write $\mathcal{P}_G$ and $\mathcal{P}_{G^\perp}$ for the projections in $L_2$ and $\mathcal{H}$ onto $G$ and its orthogonal complement $G^\perp$, respectively.

For brevity, we denote by $P_n$ the empirical measure given by the $n$ independent samples of the variables $(X, \xi)$, i.e., $P_n w = \frac{1}{n} \sum_{i=1}^n w(X_i, \xi_i)$. For example, if $h: S \to \mathbf{R}$ is a function, $P_n h^2 = \frac{1}{n} \sum_{i=1}^n h^2(X_i)$, and $P_n \xi h = \frac{1}{n} \sum_{i=1}^n \xi_i h(X_i)$.

We use the following lemmas in our proof of Theorem 1:

**Lemma 3.** *Let $\delta \in (0, 1)$. If*

$$n \geq \max\{7, 3\gamma'\} K_p \log \frac{\max\{2, 4\gamma\} p}{\delta},$$

*then, with probability at least $1 - \delta$,*

$$P_n f^2 \geq \frac{1}{2} \|f\|_{L_2}^2 - 3\sqrt{t_{p+1}} \|f\|_{L_2} \|f\|_{\mathcal{H}}$$

*for all $f \in \mathcal{H}$.*

**Lemma 4.** *There is a universal constant $C$ such that, if*

$$\frac{n}{\log^2 n} \geq C(1 \vee \gamma') \frac{K_p}{p} \frac{\|\xi\|_{\psi_1}^2}{\sigma^2},$$

*then, with probability at least $1 - \delta$,*

$$|P_n \xi f| \leq \frac{3}{2} \sigma \cdot \left( \frac{\sqrt{p} + 2\sqrt{\log 4/\delta}}{\sqrt{n}} \|f\|_{L_2} + \frac{\sqrt{\operatorname{tr} T_{G^\perp}} + 2\sqrt{t_{p+1} \log 4/\delta}}{\sqrt{n}} \|f\|_{\mathcal{H}} \right)$$

$$\leq \frac{3}{2} \sigma \cdot \left( \frac{\sqrt{p} + 2\sqrt{\log 4/\delta}}{\sqrt{n}} \right) \left( \|f\|_{L_2} + \sqrt{\gamma t_{p+1}} \|f\|_{\mathcal{H}} \right).$$

*for all $f \in \mathcal{H}$.*

With these, we prove the main result:

*Proof of Theorem 1.* We write our objective function as

$$F(f) = \frac{1}{n} \sum_{i=1}^n (Y_i - f(X_i))^2 + \alpha \|f\|_{\mathcal{H}}^2.$$

$\hat{f}$ satisfies $\nabla F(\hat{f}) = 0$. Noting that

$$\frac{1}{2} \nabla F(f) = -\frac{1}{n} \sum_{i=1}^n (Y_i - f(X_i)) k(\cdot, X_i) + \alpha f,$$

we have

$$0 = \frac{1}{2} \langle \nabla F(\hat{f}), f^* - \hat{f} \rangle_{\mathcal{H}}$$

$$= \left\langle \alpha \hat{f} - \frac{1}{n} \sum_{i=1}^n (Y_i - \hat{f}(X_i)) k(\cdot, X_i), f^* - \hat{f} \right\rangle_{\mathcal{H}}$$

$$= \alpha \langle \hat{f}, f^* - \hat{f} \rangle_{\mathcal{H}} - \frac{1}{n} \sum_{i=1}^n (Y_i - \hat{f}(X_i))(f^*(X_i) - \hat{f}(X_i))$$

$$= \alpha \langle \hat{f}, f^* - \hat{f} \rangle_{\mathcal{H}} + \frac{1}{n} \sum_{i=1}^n \left[ (Y_i - f^*(X_i))(\hat{f}(X_i) - f^*(X_i)) - (\hat{f}(X_i) - f^*(X_i))^2 \right]$$

$$= \alpha \langle \hat{f}, f^* - \hat{f} \rangle_{\mathcal{H}} + P_n \xi(\hat{f} - f^*) - P_n(\hat{f} - f^*)^2. \tag{4}$$

Let $E_1$ and $E_2$ denote the events of Lemmas 3 and 4. For part 1 of the theorem, we assume that $E_1$ holds, which occurs with probability at least $1 - \delta$. For part 2, we assume $E_1 \cap E_2$ holds, which occurs with probability at least $1 - 2\delta$. In what follows, we treat the two cases the same (and assume $\alpha > 0$), since we can simply take $\sigma = 0$ and the limit $\alpha \downarrow 0$ for part 1.

Let $e_2 = \|\hat{f} - f^*\|_{L_2}$ and $e_{\mathcal{H}} = \|\hat{f} - f^*\|_{\mathcal{H}}$. On $E_1 \cap E_2$, (4) implies

$$\frac{1}{2}e_2^2 \leq \sigma(ae_2 + be_{\mathcal{H}}) + ce_2e_{\mathcal{H}} + \alpha\langle\hat{f}, f^* - \hat{f}\rangle_{\mathcal{H}},$$

where $a = \frac{3}{2}\frac{\sqrt{p}+2\sqrt{\log 4/\delta}}{\sqrt{n}}$, $b = \sqrt{\gamma t_{p+1}}a$, and $c = 3\sqrt{t_{p+1}}$. First, note that

$$\langle\hat{f}, f^* - \hat{f}\rangle_{\mathcal{H}} = \langle f^*, f^* - \hat{f}\rangle_{\mathcal{H}} - e_{\mathcal{H}}^2 \leq \|f^*\|_{\mathcal{H}}e_{\mathcal{H}} - e_{\mathcal{H}}^2,$$

so

$$\sigma be_{\mathcal{H}} + \alpha\langle\hat{f}, f^* - \hat{f}\rangle_{\mathcal{H}} \leq (\sigma b + \alpha\|f^*\|_{\mathcal{H}})e_{\mathcal{H}} - \alpha e_{\mathcal{H}}^2 \leq \frac{(\sigma b + \alpha\|f^*\|_{\mathcal{H}})^2}{\alpha}.$$

To control the error term $ce_2e_{\mathcal{H}}$, we need a more explicit bound on $e_{\mathcal{H}}$. Because $P_n(\hat{f} - f^*)^2 \geq 0$, (4) gives

$$e_{\mathcal{H}}^2 \leq \|f^*\|_{\mathcal{H}}e_{\mathcal{H}} + \frac{1}{\alpha}P_n\xi(\hat{f} - f^*) \leq \|f^*\|_{\mathcal{H}}e_{\mathcal{H}} + \frac{\sigma}{\alpha}(ae_2 + be_{\mathcal{H}}).$$

Because $x^2 \leq a + bx$ implies $x \leq \sqrt{a} + b$, we then have

$$e_{\mathcal{H}} \leq \|f^*\|_{\mathcal{H}} + \frac{\sigma b}{\alpha} + \sqrt{\frac{\sigma a e_2}{\alpha}}.$$

Putting everything together, we have

$$\frac{1}{2}e_2^2 \leq \frac{(\sigma b + \alpha\|f^*\|_{\mathcal{H}})^2}{\alpha} + \sigma ae_2 + ce_2\left(\|f^*\|_{\mathcal{H}} + \frac{\sigma b}{\alpha} + \sqrt{\frac{\sigma a e_2}{\alpha}}\right).$$

$x^2 \leq a + bx + cx^{3/2}$ implies $x \leq \sqrt{a} + b + c^2$, so

$$e_2 \leq \sqrt{2}\frac{\sigma b}{\sqrt{\alpha}} + \sqrt{2\alpha}\|f^*\|_{\mathcal{H}} + 2\sigma a + 2c\|f^*\|_{\mathcal{H}} + 2\frac{\sigma cb}{\alpha} + 4\frac{c^2\sigma a}{\alpha}$$

$$= (\sqrt{2\alpha} + 2c)\|f^*\|_{\mathcal{H}} + 2\sigma\left(a + \frac{b}{\sqrt{2\alpha}} + \frac{bc}{\alpha} + 2\frac{ac^2}{\alpha}\right).$$

The result immediately follows by substituting our choices of $a$, $b$, and $c$ and, if $\sigma \neq 0$, using the assumption that $\alpha \geq 54t_{p+1}$. $\qquad\square$

## A.1 Proofs of key lemmas

Lemma 3 follows quickly from the following two concentration results:

**Lemma 5.** *If $\delta \in (0, 1)$, and $n \geq 7K_p\log\frac{p}{\delta}$, then, with probability at least $1 - \delta$, for all $f \in G$,*

$$P_nf^2 \geq \frac{1}{2}\|f\|_{L_2}^2.$$

*Proof.* Note that for all $f \in G$,

$$P_nf^2 = \frac{1}{n}\sum_{i=1}^{n}f^2(X_i)$$

$$= \frac{1}{n}\sum_{i=1}^{n}\langle Z(X_i), f\rangle_{L_2}^2$$

$$= \langle(P_n(Z \otimes_{L_2} Z))f, f\rangle_{L_2},$$

where we define $Z(X) = \sum_{\ell=1}^{p} v_\ell(X) v_\ell \in G$. The lemma will follow from a concentration result on $P_n(Z \otimes_{L_2} Z)$. Note that the operator $Z(X) \otimes_{L_2} Z(X) \succeq 0$ for all $X$, and, by Assumption 2, we have

$$\|Z(X) \otimes_{L_2} Z(X)\|_{L_2} = \|Z(X)\|_{L_2}^2 = \sum_{\ell=1}^{p} v_\ell^2(X) \leq K_p$$

almost surely. Also, $\mathbf{E}\, P_n(Z \otimes_{L_2} Z) = \mathbf{E}\, Z(X) \otimes_{L_2} Z(X) = \mathcal{I}_G$. The matrix Chernoff bound [1, Theorem 5.1.1] implies that, for all $\epsilon \in [0, 1)$,

$$\mathbf{P}(P_n(Z \otimes_{L_2} Z) \succeq (1 - \epsilon)\mathcal{I}_G) \geq 1 - p\left(\frac{e^{-\epsilon}}{(1 - \epsilon)^{1-\epsilon}}\right)^{n/K_p}.$$

Choosing $\epsilon = 1/2$ gives the result. $\qquad\square$

**Lemma 6.** *If $\delta \in (0, 1)$, and $n \geq \frac{3R_p}{t_{p+1}} \log \frac{2 \operatorname{tr} T_{G^\perp}}{t_{p+1}\delta}$, then, with probability at least $1 - \delta$, for all $f \in G^\perp$,*

$$P_n f^2 \leq 2t_{p+1}\|f\|_{\mathcal{H}}^2.$$

*Proof.* Similarly to the proof of Lemma 5, for all $f \in G^\perp$,

$$P_n f^2 = \langle (P_n(W \otimes_{\mathcal{H}} W))f, f\rangle_{\mathcal{H}},$$

where $W(X) = \sum_{\ell>p} t_\ell v_\ell(X) v_\ell$. Note that $\mathbf{E}\, W(X) \otimes_{\mathcal{H}} W(X) = T_{G^\perp}$. By Assumption 2,

$$\|W(X) \otimes_{\mathcal{H}} W(X)\|_{\mathcal{H}} = \|W(X)\|_{\mathcal{H}}^2 = \sum_{\ell>p} t_\ell v_\ell^2(X) \leq R_p$$

almost surely. By [1, Theorem 7.2.1], if $\epsilon \geq R_p/nt_{p+1}$, then

$$\mathbf{P}(\|P_n(W \otimes_{\mathcal{H}} W)\|_{\mathcal{H}} \leq (1 + \epsilon)t_{p+1}) \geq 1 - 2d_p\left(\frac{e^{\epsilon}}{(1 + \epsilon)^{1+\epsilon}}\right)^{nt_{p+1}/R_p},$$

where $d_p = \operatorname{tr} T_{G^\perp}/t_{p+1}$. Choosing $\epsilon = 1$ gives the result. $\qquad\square$

*Proof of Lemma 3.* Applying Lemmas 5 and 6 (with $\delta/2$ substituted for $\delta$) and a union bound, we have, with probability at least $1 - \delta$,

$$\sqrt{P_n f^2} \geq \sqrt{P_n(\mathcal{P}_G f)^2} - \sqrt{P_n(\mathcal{P}_{G^\perp} f)^2}$$
$$\geq \frac{1}{\sqrt{2}}\|\mathcal{P}_G f\|_{L_2} - \sqrt{2t_{p+1}}\|\mathcal{P}_{G^\perp} f\|_{\mathcal{H}},$$

so

$$P_n f^2 \geq \frac{1}{2}\|\mathcal{P}_G f\|_{L_2}^2 - 2\sqrt{t_{p+1}}\|\mathcal{P}_G f\|_{L_2}\|\mathcal{P}_{G^\perp} f\|_{\mathcal{H}}$$
$$\geq \frac{1}{2}\|f\|_{L_2}^2 - \frac{1}{2}\|\mathcal{P}_{G^\perp} f\|_{L_2}^2 - 2\sqrt{t_{p+1}}\|f\|_{L_2}\|f\|_{\mathcal{H}}$$
$$\geq \frac{1}{2}\|f\|_{L_2}^2 - 3\sqrt{t_{p+1}}\|f\|_{L_2}\|f\|_{\mathcal{H}}.$$

$\qquad\square$

*Proof of Lemma 4.* Let $B_2^G$ denote the $L_2$-unit ball in $G$, and let $B_{\mathcal{H}}^{G^\perp}$ denote the $\mathcal{H}$-unit ball in $G^\perp$. Note that for all $f \in \mathcal{H}$, we have

$$f \in \|f\|_{L_2} B_2^G + \|f\|_{\mathcal{H}} B_{\mathcal{H}}^{G^\perp},$$

where the plus sign denotes Minkowski addition. Therefore, because $|P_n \xi f|$ is sublinear in $f$, it suffices to bound

$$Z_1 := \sup_{f \in B_2^G} |P_n \xi f|$$

and

$$Z_2 := \sup_{f \in B_{\mathcal{H}}^{G^\perp}} |P_n \xi f|.$$

We present a complete proof for the bound of $Z_1$; the proof for $Z_2$ is similar.

First, note that

$$\begin{aligned}
Z_1 &= \sup_{f \in B_2^G} |P_n \xi f| \\
&= \sup_{\sum_{\ell=1}^p a_\ell^2 \leq 1} \left| P_n \left( \xi \sum_{\ell=1}^p a_\ell v_\ell \right) \right| \\
&= \sup_{\sum_{\ell=1}^p a_\ell^2 \leq 1} \left| \sum_{\ell=1}^p a_\ell P_n(\xi v_\ell) \right| \\
&= \left( \sum_{\ell=1}^p P_n^2(\xi v_\ell) \right)^{1/2},
\end{aligned}$$

so

$$\mathbf{E}\, Z_1 \leq \sqrt{\mathbf{E}\, Z_1^2} = \sqrt{\sum_{\ell=1}^p \mathbf{E}\, P_n^2(\xi v_\ell)} = \sigma \sqrt{\frac{p}{n}}.$$

We also have

$$\sup_{f \in B_2^G} \sum_{i=1}^n \mathbf{E}(\xi_i f(x_i))^2 = n\sigma^2.$$

Finally, note that

$$\sup_{f \in B_2^G} \|f\|_\infty \leq \sqrt{K_p},$$

so

$$\left\| \max_i \sup_{f \in B_2^G} |\xi_i f(x_i)| \right\|_{\psi_1} \leq \sqrt{K_p} \|\xi\|_{\psi_1} \log n.$$

Let $\eta \in (0,1)$. [2, Theorem 4] (with, in the notation of that paper, $\delta = 1$) implies that, with probability at least $1 - \delta/2$,

$$Z_1 \leq \sigma \left( (1+\eta)\sqrt{\frac{p}{n}} + 2\sqrt{\frac{\log 4/\delta}{n}} \right) + \frac{C_\eta' \sqrt{K_p} \|\xi\|_{\psi_1} (\log n)(\log 12/\delta)}{n}$$

for a constant $C_\eta'$ that only depends on $\eta$. By a similar argument, we have, with probability at least $1 - \delta/2$,

$$Z_2 \leq \sigma \left( (1+\eta)\sqrt{\frac{\operatorname{tr} T_{G^\perp}}{n}} + 2\sqrt{\frac{t_{p+1} \log 4/\delta}{n}} \right) + \frac{C_\eta' \sqrt{R_p} \|\xi\|_{\psi_1} (\log n)(\log 12/\delta)}{n}.$$

Fixing $\eta \in (0, 1/2)$ and choosing a suitable constant $C$ to ensure $n$ is large enough completes the proof. $\qquad\square$

## B  Proof of heat kernel approximation (Lemma 1)

In this appendix, we prove upper and lower bounds on the heat kernel diagonal values. Although we only use the upper bound in our paper, we include the lower bound also as both may be of independent interest.

The concepts from differential geometry used in this section can be found in, for example, [3, 4]. The key tools we will use in our analysis of how well the heat kernel is approximated by a Gaussian RBF are the following *comparison theorems*:

**Lemma 7** ([5, Theorem 4.5.1]). *If the sectional curvature of an $m$-dimensional manifold $\mathcal{M}$ is bounded above by $K > 0$, then, for all $x, y \in \mathcal{M}$, $k_t^{\mathrm{h}}(x, y) \le k_t^{h,K}(d_{\mathcal{M}}(x, y))$, where $k_t^{h,K}(r)$ is the (radially symmetric) heat kernel on the $m$-dimensional space of constant curvature $K$, and, if $K > 0$, we set $k_t^{h,K}(r) = k_t^{h,K}(\pi/\sqrt{K})$ for $r \ge \pi/\sqrt{K}$.*

**Lemma 8** ([5, Theorem 4.5.2]). *If the Ricci curvature of $\mathcal{M}$ is bounded below by $(m-1)K$ for some constant $K$, then, for all $x, y \in \mathcal{M}$, $k_t^{\mathrm{h}}(x, y) \ge k_t^{h,K}(d_{\mathcal{M}}(x, y))$, where $k_t^{h,K}(r)$ is the heat kernel on the space of constant curvature $K$.*

A lower bound of $K$ on sectional curvature implies a lower bound of $(m-1)K$ on the Ricci curvature tensor (see, e.g., the formula for $\mathrm{Ric}(v, v)$ in [4, p. 38]), so Lemma 8 also holds under the (stronger) assumption of a lower bound of $K$ on sectional curvature.

The space of constant curvature $K > 0$ is the sphere $S_K^m = S^m/\sqrt{K}$, while the space of constant curvature $-K < 0$ is the scaled hyperbolic space $H_K^m = H^m/\sqrt{K}$. To apply Lemmas 7 and 8, we need to find bounds for the heat kernel on the sphere and on hyperbolic space.

We will use the following result:

**Lemma 9** ([6, Theorem 1]). *The heat kernel in hyperbolic space $H^m$ has the radial representation*

$$
k_t^{h,H^m}(r) = e^{-\frac{(m-1)^2 t}{8}} \left( \frac{r}{\sinh r} \right)^{\frac{m-1}{2}} \frac{e^{-r^2/2t}}{(2\pi t)^{m/2}}
$$
$$
\times \mathbf{E}_r \exp\left( -\frac{(m-1)(m-3)}{8} \int_0^t \left( \frac{1}{\sinh^2 R_s} - \frac{1}{R_s^2} \right) ds \right),
$$

*where $R_s$ is an $m$-dimensional Bessel process, and $\mathbf{E}_r$ denotes expectation conditioned on $R_t = r$.*

A nearly identical argument to that in [6] gives a corresponding result for the sphere $S^m$ for $m \ge 2$:

**Lemma 10.** *For all $m \ge 2$, the heat kernel on the sphere $S^m$ has the radial representation*

$$
k_t^{h,S^m}(r) = e^{\frac{(m-1)^2 t}{8}} \left( \frac{r}{\sin r} \right)^{\frac{m-1}{2}} \frac{e^{-r^2/2t}}{(2\pi t)^{m/2}}
$$
$$
\times \mathbf{E}_r \exp\left( -\frac{(m-1)(m-3)}{8} \int_0^t \left( \frac{1}{\sin^2 R_s} - \frac{1}{R_s^2} \right) ds \right),
$$

*where, again, $R_s$ is an $m$-dimensional Bessel process, and $\mathbf{E}_r$ denotes expectation conditioned on $R_t = r$.*

For $m \ge 3$, the exponent in the integrands in the formula of Lemma 9 (resp. Lemma 10) is always positive (resp. negative), so we have the following simple bounds on the heat kernels on the standard spaces of constant curvature:

$$
k_t^{h,H^m}(r) \ge e^{-\frac{(m-1)^2 t}{8}} \left( \frac{r}{\sinh r} \right)^{\frac{m-1}{2}} \frac{e^{-r^2/2t}}{(2\pi t)^{m/2}}, \tag{5}
$$

and

$$
k_t^{h,S^m}(r) \le e^{\frac{(m-1)^2 t}{8}} \left( \frac{r}{\sin r} \right)^{\frac{m-1}{2}} \frac{e^{-r^2/2t}}{(2\pi t)^{m/2}}. \tag{6}
$$

It is easily verified that $p_t^{S_K^m}(r) = p_{Kt}^{S^m}(\sqrt{K}r)$, with a similar formula for scaled hyperbolic space. We can summarize this in the following result:

**Lemma 11.** *Suppose $\mathcal{M}$ is an $m$-dimensional complete Riemannian manifold for $m \ge 3$.*

1. *Suppose $\mathcal{M}$ has Ricci curvature bounded below by $-(m-1)K_1$. Then, for all $x, y \in \mathcal{M}$, denoting $r = d(x, y)$,*

$$
k_t^{\mathrm{h}}(x, y) \ge e^{-\frac{(m-1)^2}{8} K_1 t} \left( \frac{\sqrt{K_1} r}{\sinh(\sqrt{K_1} r)} \right)^{\frac{m-1}{2}} \frac{e^{-r^2/2t}}{(2\pi t)^{m/2}}.
$$

2. *Suppose $\mathcal{M}$ has sectional curvature bounded above by $K_2$. Then, for $r < \pi/\sqrt{K_2}$, and for all $x, y \in \mathcal{M}$ such that $d(x, y) \geq r$,*

$$k_t^{\mathrm{h}}(x, y) \leq e^{\frac{(m-1)^2}{8}K_2 t}\left(\frac{\sqrt{K_2}r}{\sin(\sqrt{K_2}r)}\right)^{\frac{m-1}{2}}\frac{e^{-r^2/2t}}{(2\pi t)^{m/2}}.$$

We note that, for $r = 0$ and $t$ small, these results are comparable to the well-known asymptotic expansion for the heat kernel, which depends on the scalar curvature at $x$ (see, e.g., [7, Section VI.4]).

Finally, we specialize to the case $r = 0$ and simplify:

**Proposition 1.** *Let $\epsilon \leq 2/3$.*

1. *Under the conditions of Lemma 11.1, for $t \leq \frac{8\epsilon}{(m-1)^2 K_1}$ and all $x \in \mathcal{M}$,*

$$k_t^{\mathrm{h}}(x, x) \geq \frac{1 - \epsilon}{(2\pi t)^{m/2}}.$$

2. *Under the conditions of Lemma 11.2, for $t \leq \frac{6\epsilon}{(m-1)^2 K_2}$ and all $x \in \mathcal{M}$,*

$$k_t^{\mathrm{h}}(x, x) \leq \frac{1 + \epsilon}{(2\pi t)^{m/2}}.$$

*Proof.* From Lemma 11, we have

$$e^{-\frac{(m-1)^2}{8}K_1 t} \leq (2\pi t)^{-m/2} k_t^{\mathrm{h}}(x, x) \leq e^{\frac{(m-1)^2}{8}K_2 t}.$$

The result follows from noting that $e^{-s} \geq 1 - s$ for all $s \geq 0$, and $e^s \leq 1 + \frac{4}{3}s$ for $0 \leq s \leq 1/2$. $\square$

Lemma 1 is a case of this last result, taking $K_2 = \kappa$.

## C  Proof of non-asymptotic Weyl law estimates (Theorem 2 and Lemma 2)

*Proof of Theorem 2.* By Lemma 1, for all $\lambda \geq 0$ and $t \leq \frac{6\epsilon}{(m-1)^2\kappa}$,

$$e^{-\lambda t/2}N_x(\lambda) = e^{-\lambda t/2}\sum_{\lambda_\ell \leq \lambda} v_\ell^2(x)$$

$$\leq \sum_{\ell=0}^{\infty} e^{-\lambda_\ell t/2} v_\ell^2(x)$$

$$= k_t^{\mathrm{h}}(x, x)$$

$$\leq \frac{1 + \epsilon}{(2\pi t)^{m/2}}.$$

Taking $t = m/\lambda$, we get

$$N_x(\lambda) \leq \frac{(1 + \epsilon)e^{\lambda t/2}}{(2\pi t)^{m/2}}$$

$$= \frac{1 + \epsilon}{(4\pi)^{m/2}}\frac{e^{m/2}}{(m/2)^{m/2}}\lambda^{m/2}$$

$$\leq \frac{1 + \epsilon}{(4\pi)^{m/2}}\frac{2\sqrt{m}}{\Gamma\left(\frac{m}{2} + 1\right)}\lambda^{m/2}$$

$$= \frac{2(1 + \epsilon)\sqrt{m}}{(2\pi)^m}V_m\lambda^{m/2},$$

where the second inequality uses Stirling's approximation. $\square$

*Proof of Lemma 2.* For $c \in (0,1)$, note that

$$
\begin{aligned}
\sum_{\lambda_\ell \geq \lambda} e^{-\lambda_\ell t/2} v_\ell^2(x) &\leq e^{-(1-c)\lambda t/2} \sum_{\lambda_\ell \geq \lambda} e^{-c\lambda_\ell t/2} v_\ell^2(x) \\
&\leq e^{-(1-c)\lambda t/2} \sum_{k=0}^{\infty} e^{-c\lambda_\ell t/2} v_\ell^2(x) \\
&= e^{-(1-c)\lambda t/2} p_{ct}^{\mathcal{M}}(x,x) \\
&\leq e^{-\lambda t/2}(1+\epsilon) \frac{e^{c\lambda t/2}}{(2\pi ct)^{m/2}}.
\end{aligned}
$$

Choosing $c = m/\lambda t$, the remainder of the proof is identical to that of Theorem 2. $\qquad\square$

## D    Proof of manifold regression results (Theorems 3 and 4)

*Proof of Theorems 3 and 4.* To apply the framework of Sections 2.1 and 4.1, which assumes the set $S$ has measure 1, we consider the normalized volume measure $d\widetilde{V} = dV/\operatorname{vol}\mathcal{M}$. With respect to $\widetilde{V}$, $k_t^{\mathrm{h}}$ has the eigenvalue decomposition

$$
k_t^{\mathrm{h}}(x,y) = \frac{1}{\operatorname{vol}\mathcal{M}} \sum_\ell e^{-\lambda_\ell t/2} \tilde{u}_\ell(x)\tilde{u}_\ell(y),
$$

where $\tilde{u}_\ell = \sqrt{\operatorname{vol}\mathcal{M}} u_\ell$. A similar normalized expansion holds for $k_\Omega^{\mathrm{bl}}$.

Note that Theorem 2 and Lemma 2 only give us bounds on the contants $K_p$ and $R_p$ in Assumption 2. For $k_\Omega^{\mathrm{bl}}$, this holds with $K_p = p(\Omega)$ (taking $\epsilon = 1/2$ in Theorem 2) and $R_p = 0$. Assumption 3 holds trivially with $\gamma = \gamma' = 0$.

For $k_t^{\mathrm{h}}$, we can again take $K_p = p(\Omega)$ (again taking $\epsilon = 1/2$), and we get a bound on $R_p$ such that $\gamma = \gamma' = 1$.

Finally, for both kernels, we take into account the fact that $\|\cdot\|_{L_2(\mathcal{M},\widetilde{V})} = \|\cdot\|_{L_2(\mathcal{M},V)}/\sqrt{\operatorname{vol}\mathcal{M}}$. With these considerations in mind, the results follow from Theorem 1. $\qquad\square$