[Reviews · NeurIPS 2020]

Review 1

Summary and Contributions: The paper provides an analysis of kernel ridge regression on a manifold. In particular, it first computes the effective dimension associated to the manifold with Haar measure and then it uses this result to compute kernel ridge regression.

Strengths: The paper provides an explicit non-asymptotic bounds on the affective dimension of the a manifold with Haar measure.

Weaknesses: The paper provides an explicit non-asymptotic bounds on the affective dimension of the a manifold with Haar measure with the very strong assumption that the considered kernel is based on the Laplace-Beltrami operator of the Manifold, e.g. k = e^(-t \Delta_M), where \Delta_M is a Laplace-Beltrami operator. The papers [49], [40] and also the more classical "Optimal rates for the regularized least-squares algorithm" De Vito, FoCM 2007 determines the optimal rates of kernel ridge regression for a generic kernel, input domain (beyond manifolds) and probability distribution on the data, but given the effective dimension of the problem. Then a result analogous to Theorem 1 can be obtained by plugging Theorem 2 in one of the papers listed above. This comment does not necessarily diminish the impact of the paper. However I would make it clear in the discussion of Theorem 1 and I would also clarify what the proposed result improves.

Correctness: The claims and methods are correct

Clarity: The paper is well written

Relation to Prior Work: See weakness section

Reproducibility: Yes

Additional Feedback:


Review 2

Summary and Contributions: The authors consider the theory of regression on a manifold using reproducing kernel Hilbert space methods. The paper establishes a novel nonasymptotic version of the Weyl law from differential geometry. The authors show that a kernel regression estimator yields error bounds controlled by the effective dimension. Contributions: The authors show that some certain spaces of smooth functions on a manifold are effectively finite-dimensional with complexity scaling to manifold dimension rather than ambient dimension.

Strengths: The authors derived nonasymptotic error bounds for general regression and interpolation in an RKHS via Weyl law from differential geometry and then establish sample complexity bounds for regression on a manifold with the help of heat kernel.

Weaknesses: The authors do not propose a new algorithm in this paper but only establish the theoretical results to reveal the relationship between the intrinsic dimension and the regression on manifolds. The function space concerned is the classic RKHS space using the heat kernel. It’s unclear to me whether the author's theoretical results have significantly improved the previous regression or classification algorithms. The author did not design further experiments to illustrate this point. The innovation and contribution of the article are limited and unattractive. The authors derive the upper error bound for regression in RHKS when the data is distributed on low-dimensional manifolds. I wonder how the intrinsic dimension of the data is determined. Do high-dimensional data always have a corresponding low dimensional manifold representation? Can it be verified from theory or experiments? There have been many theoretical works on the regression on manifolds, such as the work 24 cited in Section 3.2. which shows that if the function has a bounded derivative, the minimax optimal convergence rate can be guaranteed. In this paper, the authors use the kernel function to get similar results too, except that the constant in front of the convergence rate was improved. The theoretical innovation is not enough. In section 4.2, the authors derive the error bounds of the function regression on the manifold with the help of heat kernel. However, the authors do not provide an effective algorithm or designed experiments to verify the theoretical results. Since the author does not provide an effective algorithm that can be implemented in practice, results in this paper seems to have no practical value.

Correctness: I managed to check the important points and calculation carefully. I think both claims and method are correct.

Clarity: This paper is well written and the proves are clear and well organized. The authors provide all details of the proof in the supplementary material.

Relation to Prior Work: It is unclearly discussed how this work differs from previous contributions. In the third Section ‘related work’, the author mention the method of manifold dimensionality reduction, and the application of high-dimensional statistics on low-dimensional data models, both scarcely related to the problems discussed in the paper. In sections 3.2 and 3.3, The author mention the previous work related to manifold regression, and only illustrate that the theoretical results improve the constant in front of the optimal convergence rate. I don’t think the authors clearly discuss how this work differs from previous contributions.

Reproducibility: Yes

Additional Feedback: I suggest that the author propose a new regression algorithm on the basis of the theoretical results in this paper. This will not only facilitate the further design of experiments to verify the rationality of the theoretical results, but also make the results of this paper more useful in real applications. First of all, I’d like to see the authors discuss how to find the effective dimension of real data based on the existing theoretical results. Does such an effective dimension always exist? Secondly, I suggest the authors to explore whether such theoretical results can improve the performance of the previous algorithms. I think these would help a lot to make the work more meaningful, enlightening and practical.


Review 3

Summary and Contributions: This paper studies RKHS ridge regression when the feature space is a manifold of upper bounded curvature. The sample complexity depends on the dimension of the manifold, instead of that of the ambient space. This dependency is obtained via a new nonasymptotic version of Weil's inequality for the heat kernel on smooth maniforlds, which is of independent interest.

Strengths: The results on RKHS ridge regression are pretty standard, but the main strength of this paper is the use of a new upper bounds for the heat kernel on smooth manifolds with bounded curvature from above. I believe that this work will be relevant to the NeurIPS community.

Weaknesses: In my opinion, this paper has a couple of weaknesses. 1. The main weakness, in my opinion, is that this work requires the knowledge of the manifold. Otherwise, the ambient dimension will appear in the rates, but this is not considered in this work. However, in that case, with the improved Weil's inequality in hand (which is truly the only important result in this work), everything can follow pretty easily from previous works on learning with RKHS. 2. It does not discuss optimality of the results. In particular, the dependency on $p$, $\Omega$ and $m$ should be discussed in further details. In addition, under some extra smoothness assumptions on $f^*$, can a tradeoff be achieved between the bias and the variance terms in Theorem 4? 3. It only considers manifold with curvature bounded from above, by a positive constant. What happens when this constant is zero (the manifold is then NPC, which is a relevant case to consider) or when it is negative? What happens when the curvature is bounded from below? The latter is discussed in some lemmas, but not in the statistical results.

Correctness: All the claims and proofs seem correct to me.

Clarity: The paper is well written. Just a few remarks below. 1. Line 230, there should be f^* instead of f 2. In Theorem 1.1 (Line 231), the author should recall that alpha is the regularizing parameter. In addition, under the absence of noise, it seems a bit strange to write a result for alpha \neq 0. 3. Line 469: There should be \mathcal H instead of H (twice). 4. Lemma 8: Why consider curvature bounded from below here, whereas the learning results do not cover this case?

Relation to Prior Work: The related literature seems fairly well discussed in this work.

Reproducibility: Yes

Additional Feedback:


Review 4

Summary and Contributions: This paper gives a bound of approximated function for the problem of regression using RKHS, when the samples are uniform on a Riemannian manifold. Furthermore, using the effective finite-dimension, they generalized the non-asymptotic version of Weyl law.

Strengths: Theoretically well-developed.

Weaknesses: Major Comments: 1. The authors proposed sample complexity for regression problem using RKHS. Though the theory seems solid and statement of the theorems make sense (didn't go through the detailed proof), the paper seems "dry". Some analysis of the theory are missing, e.g. (a) The main theorem, theorem 4 gives the bound on the function approximation error if the function has values uniformly over the manifold. In most practical cases this is not the case, hence the analysis of how crucial the constraint is to get the bound is essential. (b) In Lemma 1, in the satement, the author should comment on what happens if the sectional curvature is not bounded. For example, for homogeneous spaces, the sectional curvature is upper bounded but in the case of a general manifold, \kappa may not be finite. (c) The statement of the Theorem 3 does not make sense. It ends with "Let \hat{f} be the kernel regression estimaye with kernel ..." I don't understand what the theorem 3 trying to prove. (d) In theorem 4, for some value of p as defined in Eq. (3), we can have a \delta very close to 0, in that case, the right hand side of the upper bound on the norm of the approximation error between \hat{f} and f^* blows up and the bound becomes trivial. Is there any way to prevent that? Or can the authors comment on the implications on the choice of p. (e) Although I enjoy reading the paper, I don't see much further development for the NeurIPS community generated from this paper. May be a better venue is a Differential Geometry journal. Additional Comments: 1. Number the equations.

Correctness: The theorems are technically correct, although I didn't check the proofs presented in the supplement.

Clarity: Very well written.

Relation to Prior Work: Yes

Reproducibility: No

Additional Feedback: Although it does not have much application now, it is a very solid theoretical piece. Hence, I recommend acceptance.

[Author Response · NeurIPS 2020]

We appreciate the reviewers' attention and thoughtful feedback. We are grateful that there was a consensus that our novel (nonasymptotic) Weyl law and its implications for kernel regression on a manifold represent an important theoretical contribution. Below, we respond to the major themes of the reviews. Several reviewers also provided more technical comments that, while we do not address them here, will be incorporated into our final version of the paper.

**Practical relevance**. We certainly acknowledge that we may not know the exact heat kernel in many practical situations. While there are means of estimating it from data, our treatment assumes that this is known *a priori*. We consider this paper to be an initial attempt to determine what is possible under ideal circumstances; methods which use alternative fixed or estimated kernels may not do as well, but we provide guidance as to what one might hope to achieve. We believe this provides a firm foundation for future work on manifold function estimation that may be able to address this gap.

**Relation to prior work**. As some reviewers noted (and as we mention in lines 185–190 and 244–249), our general result on kernel regression (Theorem 1) is similar to previous results such as those in [49] and [50], although our result has the advantage of being applicable to the noiseless case without regularization.

Much of the prior work, such as that in [35] (Caponetto and De Vito) and [40], primarily considers what power-law error rate (i.e., $\|\hat{f} - f^*\|_{L_2}^2 \lesssim n^{-\gamma}$) we can obtain if we are given power-law behavior of the kernel integral operator eigenvalues $t_\ell$ (or, equivalently, the regularized effective dimension $\mathrm{tr}(\mathcal{T} + \lambda \mathcal{I})^{-1} \mathcal{T}$ as $\lambda \downarrow 0$). Our result is **nonasymptotic** (unlike [35]) and provides **explicit constants** (unlike [35] and [40]). Furthermore, it can be applied to more general kinds of eigenvalue decay (such as the exponential decay that we find for the heat kernel). In the power-law case, when $t_\ell \approx \ell^{-b}$, an optimal choice of the dimension $p$ in the bias and variance terms of our result recovers the standard minimax optimal error rate $\|\hat{f} - f^*\|_{L_2}^2 \lesssim n^{-b/(b+1)}$ that was proved in [35].

A key feature of our work is that the functions we consider are *much smoother* than in the classical models that many of the references consider (e.g., belonging to a Sobolev space, or, similarly, having bounded $m^{\text{th}}$ derivative as in [24], which both lead to power-law error rates). Our results show that this smoothness is indeed exploited, as we obtain finite-dimensional error rates (within a logarithmic factor, in the heat kernel case), and the constants in the error and the sample complexity correspond clearly to the finite effective dimension of the function space that we analyze.

**Optimality**. Our results are certainly optimal (within multiplicative constants) in the case of bandlimited functions. It is well-established that to estimate a function in a $p$-dimensional subspace $G$, the (squared) $L_2$ error due to noise will always be of the order $p/n$, where $n$ is the number of samples. The sample complexity $n \approx p \log p$ is also optimal, where $p \approx \Omega^m$ and $m$ is the manifold dimension: considering the simple example of the torus $T^m$, estimating $\Omega$-bandlimited functions requires that every point in the domain is within $O(1/\Omega)$ of a sample point; considering a grid over the domain (which has $O(\Omega^m)$ regions), the coupon collector problem suggests that we will need $O(\Omega^m \log \Omega^m) \approx p \log p$ points sampled uniformly at random to achieve a sufficiently fine sampling.

**Additional comments**. We assume in our paper that we sample points uniformly over the manifold. As we mention in lines 250–254, it might be an interesting extension to consider other sampling measures. However, the uniform sampling assumption requires the manifold to be compact. We also assume, for simplicity, that the manifold is without boundary (although much of this work could be extended to the case with boundary).

Regarding R3's $3^{\text{rd}}$ question, our key upper bound on the heat kernel (Lemma 7 in Appendix B) only applies to *positive* upper bounds on sectional curvature. We agree with R3 that it would be very interesting to investigate what happens with a nonpositive upper bound. Regarding the second part of the question, since our nonasymptotic Weyl law (Theorem 2) is only an *upper* bound, its proof in Appendix C only requires an upper bound on the heat kernel, which in turn only requires an upper bound on sectional curvature. A corresponding lower bound in the Weyl law would likely require a corresponding lower bound on the curvature.

Regarding R4's question 1(b), the compactness assumption also ensures that the sectional curvature is uniformly (upper) bounded over the manifold.

Regarding R4's concern regarding the statement of Theorem 3, please see the continuation of the theorem statement onto the next page.

Finally, to clarify the comment on Theorem 4 in lines 291–292, this is intended to be an "alternative interpretation" of the theorem, where we choose a tolerance $\delta$ for the bias term. Because the $p$ defined in line 292 only depends logarithmically on $\delta$, we can choose $\delta$ to be very small without greatly affecting the "effective dimension" $p$.

[Meta-Review · NeurIPS 2020]

The reviewer consensus is that it is a solid (if somewhat dry) theoretical contribution, lauding in particular the intrinsic interest of the Weyl-Type inequality.